# Combining Crude Glycerin with Chitosan Can Manipulate In Vitro Ruminal Efficiency and Inhibit Methane Synthesis

**DOI:** 10.3390/ani10010037

**Published:** 2019-12-23

**Authors:** Anuthida Seankamsorn, Anusorn Cherdthong, Metha Wanapat

**Affiliations:** Tropical Feed Resources Research and Development Center (TROFREC), Department of Animal Science, Faculty of Agriculture, Khon Kaen University, Khon Kaen 40002, Thailand; aontoranu@gmail.com (A.S.); metha@kku.ac.th (M.W.)

**Keywords:** gas kinetic, exoskeletons, rumen fermentation, propionate, greenhouse gas

## Abstract

**Simple Summary:**

Methane (CH_4_) inhibition approaches in ruminants are considered to provide economic benefits and reduce global greenhouse gases. The inclusion of crude glycerin at 21% in to total mixed ration (TMR) diets with chitosan supplementation at 2% enhanced ruminal propionate concentration and reduced methane production without causing any detrimental effect on the gas kinetics or nutrient digestibility.

**Abstract:**

It was hypothesized that the combination of glycerin and chitosan improves ruminal fermentation efficiency via an enhanced propionate (C3) and reduces in vitro CH_4_ production. This was explored through in vitro gas production with substrates containing crude glycerin, which replaced cassava chips in the studied ration. The experimental design was organized following a 3 × 3 factorial in completely randomized design and the arrangement of treatments were different levels of crude glycerin supplementations 0, 10.5, and 21% of total mixed ration (TMR) and chitosan levels were added at 0, 1, and 2% dry matter (DM) of substrate. Then, 0.5 g of TMR substrates were added into 40 mL bottles, together with respective doses of chitosan and then incubated at 39 °C. The dietary treatments were performed in three replicates within the incubation, and incubations were repeated on three separate days (runs). No interactions were found between crude glycerin and chitosan doses in terms of theoretical maximum of asymptotic gas production (*b*), rate of gas production (*c*), the discrete lag time prior to gas production (*L*), or the cumulative gas production at 96 h of incubation (*p* > 0.05). Cumulative gas production at 96 h of incubation was similar among the doses of crude glycerin and levels of chitosan, which ranged from 64.27 to 69.66 mL/g DM basis of substrate (*p* > 0.05). The concentration of ruminal NH_3_-N after 2 and 4 h of incubation ranged from 14.61 to 17.10 mg/dL and did not change with the addition of crude glycerin with chitosan (*p* > 0.05). The concentration of CH_4_ after 2 h of incubation did not change among treatments (*p* > 0.05), whereas after 4 h of incubation, CH_4_ synthesis was significantly reduced by enhancing doses of crude glycerin and chitosan (*p* < 0.05). The combination of 21% of crude glycerin in TMR with 2% chitosan depressed CH_4_ production as much as 53.67% when compared to the non-supplemented group. No significant crude glycerin and chitosan interaction effect was detected for in vitro digestibility of nutrients after incubation for 12 and 24 h using the in vitro gas production technique (*p* > 0.05). In addition, no significant changes (*p* > 0.05) were observed in total volatile fatty acids, acetate (C2) or butyrate content among treatments and between the main effects of crude glycerin with chitosan. At 4 h of incubation, ruminal C3 content and the C2 to C3 ratio changed significantly when crude glycerin and chitosan was added (*p* < 0.05). The 21% crude glycerin incorporate into TMR, in combination with 2% additional chitosan, increased C3 content by 26.41%, whereas the ratio of C2 to C3 was reduced by 31% when compared to the control group. Propionate concentration increased by 11.75% when increasing levels of chitosan at 2% of substrate, whereas the C2 to C3 ratio decreased by 13.99% compared to the 0% chitosan group. The inclusion of crude glycerin at 21% in TMR diets with chitosan supplementation at 2% enhanced ruminal propionate concentration and reduced methane production without causing any detrimental effect on the gas kinetics or nutrient digestibility.

## 1. Introduction

Methane is a major greenhouse gas synthesized during ruminal feed fermentation [1]. Methane production supplies approximately 5% to 7% of the feed’s gross energy, which constitutes approximately 16 to 26 g/kg of feed consumed [2]. Thus, CH_4_ inhibition approaches in ruminants are considered to provide economic benefits and reduce global greenhouse gases [3].

Crude glycerin is a residue from biodiesel production [4]. In the rumen, glycerin is quickly converted to propionate. Propionate serves as an H sink in the rumen, thus an increase in its concentration would reduce H availability and therefore CH_4_ production. Lee et al. [5] indicated that supplemental glycerin reduced CH_4_ formation in alfalfa and corn by 25.2% and 14.8%, respectively, thus reducing the methanogenic potential of forages, so the addition of glycerin might be an influential means of inhibiting ruminal CH_4_ formation. Similarly, van Cleef et al. [6] indicated the added possibility of glycerin decreasing CH_4_ synthesis and negatively affecting microbial synthesis and feed digestion. However, the studied results differed in that they indicated no change in CH_4_ production when supplemented with glycerin [7]. This could be due to various factors that may be responsible for inconsistent CH_4_ production results, such as the levels used, the purity of the glycerin and the in vitro substrate diets. Therefore, an evaluation of glycerin is still necessary to elucidate the specific factors to confirm their effects on ruminal CH_4_ production.

Chitosan is a biopolymer found in the exoskeletons of crab and shrimp. Interestingly, chitosan can shift volatile fatty acid (VFA) profiles and change ruminal fermentation, as well as increase C3 concentration and, consequently, reduce CH_4_ production [8]. Furthermore, CH_4_ reduction could be due to the degree of deacetylation found in the chitosan, which may modify the permeability of methanogenic bacteria cell walls [9]. A previous experiment by Goiri et al. [10] demonstrated that the addition of chitosan can inhibit in vitro ruminal CH_4_ synthesis when high concentrations are added to the substrate. Furthermore, adding chitosan at 2% of substrate resulted in a strong reduction effect on CH_4_ formation [8].

Therefore, it was hypothesized that the combination of glycerin and chitosan improves ruminal fermentation efficiency via an enhanced C3 and reduces in vitro CH_4_ production. This was explored through in vitro gas production with substrates containing crude glycerin, which replaced cassava chips in the studied ration.

## 2. Materials and Methods

Animals procedures were allowed by the Animal Ethics Committee of Khon Kaen University (protocol number 9/2561), based on the Ethic of Animal Experimentation of National Research Council of Thailand.

### 2.1. Dietary Preparation

Chitosan in the experiment was prepared using Toan’s method [11]. Briefly, fresh shrimp were collected from a local market in Khon Kaen province, Thailand and cleaned with water. The shrimp shells were allowed to achieve partial autolysis for 2 days before chitosan extraction. Shrimp shell autolysis was performed with 0.68 M HCl solution (1:5 *w*/*v*) at 26–30 °C for 16 h. The residue was rinsed and soaked in tap water for 6–8 h. It was then removed from the water, and protein was removed using 0.62 M NaOH solution (1:5 *w*/*v*) at 26–30 °C for 20 h. The chitin product was stripped of acetyl groups using NaOH at 65 °C for 20 h, after which the chitosan was obtained. The chitosan was rinsed and dried in sunlight before being used for chemical composition analysis and further investigation.

Crude glycerin used in the experiment was offered from the Specialized Research and Development Center for Alternative Energy from Palm, Prince of Songkla University, Thailand and chemical contents are presented in Table 1.

### 2.2. Mixed Rumen Inoculum

In experiments, two male, rumen-fistulated dairy cattle weighing 370 ± 50 kg of body weight were supplied as rumen fluid donnors. The animals were fed with TMR1 (control diet; Table 1) at 2.5% body weight (BW) daily. The cattle were offered clean water, in separate pens with free access to mineral blocks. The rumen fluid was collected at before feeding (06.00 am), was filtered with cheesecloth into warmed flasks and then taken to the laboratory. Before starting to prepare a batch culture, ruminal fluid from the two cattle was mixed at equal proportions. The artificial saliva was performed following to the protocol of Menke and Steingass [12] and mixed with ruminal fluid at 2:1 ratio and used for ruminal inoculum. The serum bottles were incubated in a 39 °C water bath and then added with 40 mL of mixed ruminal inoculums.

### 2.3. Design and Substrate Preparation

The experimental design was organized following a 3 × 3 factorial in completely randomized design and treatments are the combination of the 3 levels of glycerin (0, 10.5, 21% of TMR) (Table 1) and 3 levels of chitosan (0, 1 and 2% DM of substrate). The TMR diets were dried in a hot air-oven at 60 °C for 48 h, ground to pass through a 1-mm sieve and prepared for nutritional composition determination and the gas production study. Then, 0.5 g of TMR substrates were added into 40 mL bottles, to which respective doses of chitosan were added, and then incubated at 39 °C. Every 3 h, the bottles were gently shaken. The dietary treatments were tested in three replicates within the incubation, and incubations were repeated on three separate days (runs). Three bottles (no substrate) were always included with each run as blank control. A set of 54 bottles (3 bottles per treatment × 9 treatments × 2 sampling times: 2 and 4 h incubation) were separately collected for pH, ammonia-nitrogen (NH_3_-N), volatile fatty acids (VFAs) and CH_4_ determination. A digestibility evaluation was arranged with different set of 54 bottles (3 bottles per treatment × 9 treatments × 2 sampling times at 12 and 24 h incubation).

### 2.4. Chemical Analyses and Fermentation Measurements

Chemical analysis was performed on the TMR and chitosan. Dry matter (DM), ash, crude protein (CP), organic matter (OM), ether extract (EE), and acid detergent fiber (ADF) were measured according to the Association of Official Analytical Chemists (AOAC) [13]. The neutral detergent fiber (NDF) was analyzed according to the protocol of Van Soest et al. [14]. Chitosan was analyzed for CP and OM with the similar protocol mentioned above, while the degree of deacetylation and solubility were investigated first by derivative ultraviolet (UV) spectrophotometry and then the transglucosidase method modified by Toan [11].

After incubation for 0, 0.5, 1, 2, 4, 6, 8, 12, 18, 24, 48, 72, and 96 h, gas production was measured by using a pressure transducer and a calibrated syringe. Fermentation liquor was collected at 2 and 4 h of incubation to investigate the ruminal pH and then filtered through 4 layers of cheesecloth. The fermentation liquor was centrifuged at 16,000 × *g* for 15 min, and the supernatant was analyzed for NH_3_-N by Kjeltech Auto 1030 analyzer (Tecator, Hoganiis, Hoganas, Sweden). Total VFA, C2, C3, and C4 were evaluated using high-performance liquid chromatography (HPLC) (ETL Testing Laboratory, Inc., Cortland, NY, USA). In vitro digestibility was elucidated at 12 and 24 h of incubation, when the content was filtered through pre-weighed Gooch crucibles (40 mm of porosity) and residual DM was calculated. The difference in weight of DM and its components before and after incubation represented in vitro DM digestibility (IVDMD) and in vitro organic digestibility (IVOMD), in vitro NDF digestibility (IVNDF), and in vitro ADF digestibility (IVADF) [15]. Methane gas production in the total gas was evaluated at 2 and 4 h post incubation, and 10 mL of gas was withdrawn from the head space of the fermented bottle using a leak-proof syringe and analyzed for CH_4_ using gas chromatography (Shimadzu Corporation, Kyoto, Japan).

### 2.5. Statistical Analysis and Gas Kinetics

Estimates of the kinetic parameters of gas production (mL/g DM basis) were obtained using the NLIN procedure according to France et al. [16] using the following model:Gas production = *b* × [1 − *e^-c(t-L)^*]
where *b* is the asymptotic gas production (mL/1 g DM); *c* is the rate of gas production (/h) and *L* (h) is the discrete lag time prior gas production.

Data of in vitro gas kinetics, cumulative gas production, CH_4_, NH_3_-N, pH, VFAs and digestibilities were subjected to the general linear model (GLM) procedures of SAS [17]. Differences among treatment means for all parameters were contrasted by Tukey’s multiple comparison test.

## 3. Results and Discussion

### 3.1. Nutritional Content of Feed Test

The experimental diet’s formulation and nutritional content are shown in Table 1. TMR1 (control diet) contained a high level of cassava chips at 40% as the main energy source, while rice straw was used as roughage source at 30%. When replacing cassava chips with crude glycerin as a main energy source, it should be noted crude glycerin does not provide protein or essential minerals; thus, these were compensated for. Cassava chips were replaced with crude glycerin from 10.5% to 21.0% in TMR diets as a fermentable energy source. TMR3 reduced cassava chip inclusion by 50% when compared to TMR1. TMR diets contained similar CP contents at 14.05% to 14.32% DM, whereas EE content increased when the level of crude glycerin was increased (4.88% to 8.19% DM). TMR1 (control diet) contained EE at 1.03% DM. This might be because crude glycerin contains high EE levels (48.75%). The NDF content in TMRs was reduced to 40.98% DM when decreasing cassava chips by 20% in TMR diets. This is related to the cassava chips’ replacement by crude glycerin, which contains low NDF levels. Crude glycerin contains low methanol at 4.53% DM and was lower than the levels reported by Lage et al. [18], who demonstrated that 8.7% of methanol in crude glycerin did not adversely affect the health status of feedlot lambs fed crude glycerin 12% on DM basis of concentrate diet.

The removal of protein and minerals from shrimp shell, or chitin, was determined for ash and CP content to evaluate whether these critical values are lower than 1% [11]. The chitosan’s quality, such as its degree of deacetylation and solubility were measured to characterize the preparation process. The shrimp shell-based chitosan extract had a uniform quality, with 0.53% CP and 0.27% ash (Table 1). Its solubility and deacetylation were 98.70% and 88.00%, respectively, which indicates that chitosan produced from shrimp shells has potential for animal supplementation use.

### 3.2. Gas Production Profiles

No interactions were found between crude glycerin and chitosan doses in terms of theoretical maximum of asymptotic gas production (*b*), the rate of gas production (*c*), the discrete lag time prior to gas production (*L*), or the cumulative gas production at 96 h of incubation (*p* > 0.05; Table 2). It was found that the values of “*b*” did not vary among treatments (*p* > 0.05) and ranged from 66.86 to 72.25 mL/g DM basis. Cumulative gas production at 96 h of incubation was similar among the doses of crude glycerin and levels of chitosan, which ranged from 64.27 to 69.66 mL/g DM basis of substrate (*p* > 0.05). These results demonstrated that increasing doses of crude glycerin to 21% and chitosan to 2% did not adversely influence gas kinetics and cumulative gas production. This is in agreement with Avila et al. [7], who indicated that gas production did not change when glycerin was used as a substitute for barley grain up to 21% DM in a feed that included a similar ratio of barley grain and barley silage. However, Krueger et al. [19] found that the inclusion of 10%–40% of glycerin in alfalfa hay could increase gas production, which may occur because it is rapidly metabolized, supplies higher synchronism with the nitrogen sources, and accordingly yields high gas levels, but other work by Ferraro et al. [20] indicated lower gas yields from glycerin compared to those of alfalfa or corn silage. The differences among trials might be different affect gas yields.

### 3.3. In Vitro Rumen Parameters and CH_4_ Concentration

The influence of crude glycerin with chitosan on pH, NH_3_-N concentration, and CH_4_ synthesis after 2 and 4 h of incubation using an in vitro gas production test are presented in Table 3. There were no interactions on all parameters between crude glycerin and chitosan. Ruminal pH was not affected by levels of crude glycerin with chitosan (*p* > 0.05) and demonstrated a consistent range of 6.65 to 6.76. Considering it is a buffered medium which is an optimum pH for ruminal maintenance and microbial activity. The concentration of ruminal NH_3_-N after 2 and 4 h of incubation ranged from 14.61 to 17.10 mg/dL and did not change with the addition of crude glycerin with chitosan (*p* > 0.05). This could be due to the TMR diet, which was isonitrogenous and the crude glycerin with chitosan introduced no N source to the incubation (Table 1). The concentration of CH_4_ after 2 h of incubation was not different among treatments (*p* > 0.05) which might be due to the initial ruminal fermentation process, thus lowering substrate for CH_4_ to synthesis. However, after 4 h of incubation, CH_4_ synthesis was significantly reduced by enhancing doses of crude glycerin and chitosan (*p* < 0.05). 

Methane concentration was inhibited by 39.65% when 21% crude glycerin was included in TMR, compared to the TMR group with 0% crude glycerin. Regarding the influence of crude glycerin on in vitro CH_4_ synthesis, Vito et al. [21] indicated that concentration of CH_4_ could be reduced with glycerin addition because of the adverse influence of glycerin on methanogen populatios. Vito et al. [21] also suggested that supplementation of 28% of crude glycerin in Nellore steers grazing on tropical grass reduced methanogen population by 55.93% compared to the control group. This might occur because the crude glycerin can modify the cell membrane permeability of methanogens, thus inhibiting the CH_4_ production. Furthermore, Vito et al. [21] found that the linear reduction in the number of protozoa of the genera *Entodinium* and *Isotricha* when the increasing level of crude glycerin could affect CH_4_ reduction. In addition, Lee et al. [5] demonstrated that glycerin has the potential to inhibit in vitro CH_4_ synthesis in the rumen, suggesting an advantageous impact of glycerin on enhancing energy utilization. The small CH_4_ production in the present work might be ascribable in part to the small NDF concentration of the TMR and the use of crude glycerin, which replaced cassava chips [7]. Higher NDF contents in TMR1 (5.23%) resulted in greater CH_4_ production, and TMR3 produced 22.03 mL/1 g less CH_4_ compared with TMR1, which produced CH_4_ at 36.51 mL/1 g DM basis.

Supplementation of chitosan at 2% could reduce CH_4_ production by 26.05% compared to no chitosan supplementation. Similarly, a substantial decrease in CH_4_ production (~42%) was found when highly soluble chitosan (98.70% deacetylated) was added at 7% to a diet by Rusitec systems [10]. The present study also found that a similar decrease in CH_4_ synthesis without an adverse effect on diet digestion could be accomplished by reducing chitosan solubility (88.00% deacetylated). This agrees with previous reports which indicated no adverse effect of chitosan on nutrient digestibility when little roughage was used in vivo [22,23] or in vitro [9]. Moreover, Haryati et al. [8] noted that CH_4_ production decreased when chitosan was added at 2% of substrate. The mode of action by which chitosan reduces CH_4_ synthesis in ruminants has been proposed. This could be due to properties such as chitosan’s high degree of deacetylation, which may modify methanogen’s cell wall permeability, thus reducing CH_4_ production [8]. In addition, positively charged chitosan may interfere with negatively charged methanogens and cause a leakage of protein and other intracellular compositions from the cytosol [9]. An additional theory related to chitosan is that of agglutination. Chung et al. [24] proposed that chitosan has a positive charge and, at lower concentrations than 0.2 mg/mL of incubation fluid, can attach to negatively charged methanogenic surfaces to cause agglutination.

The combination of 21% of crude glycerin in TMR with 2% chitosan depressed CH_4_ production as much as 53.67% when compared to the nonsupplemented group. Interestingly, it has been reported that crude glycerin and chitosan could shift the VFA profile and change ruminal fermentation, as well as increase C3 concentration, resulting reduced CH_4_ production [5]. The fermentation of crude glycerin and chitosan potentially impacts the production of a H_2_ sink and could support in the conversion of carbohydrate fermentation from C2 to C3 production. The conversion could affect the overall electron equilibrium in the rumen and decrease the availability of hydrogen for CH_4_ synthesis. This result could confirm that the combination of crude glycerin and chitosan might reduce CH_4_ production more effectively than individual supplementation. However, the present study found CH_4_ was reduced in the early fermentation of 4 h of incubation, thus the inhibitory effect of glycerin and chitosan throughout the complete fermentation process (96 h of incubation) should be further elucidated.

### 3.4. In Vitro Digestibility

No significant crude glycerin and chitosan interaction effect was detected for in vitro digestibility of nutrients after incubation for 12 and 24 h using the in vitro gas production technique (Table 4; *p* > 0.05). Increasing levels of crude glycerin up to 21% DM in TMR diet did not change IVDMD, IVOMD, IVNDFD, or IVADFD after 12 and 24 h of incubation (*p* > 0.05). In vitro dry matter digestibility at 12 and 24 h of incubation ranged from 53.34% to 56.29% DM and 65.70% to 67.39% DM, respectively, for all levels of crude glycerin inclusion. This result indicated that crude glycerin could be a substitute energy source in TMR diets without adversely affecting feed utilization and agreed with Schröder and Südekum [25], who replaced 50% of the starch in the cows’ diet with 10% glycerin and found no adverse effect on DM intake and feed digestion. In addition, Saleem et al. [26] reported enhanced fiber digestion with supplemental glycerin. Furthermore, van Cleef et al. [6] concluded that the effective degradation and the in vitro DM digestibility of diets were improved when crossbred sheep were fed a partial or total replacement of corn cracked grain with crude glycerin at high doses. However, Abu Ghazaleh et al. [4] noted that glycerin addition decreased in vitro NDF digestion due to the decreasing bacterial counts associated with feed digestibility. Thus, the varying influence of glycerin addition on ruminal digestibility could be affected by the different levels and chemical composition of crude glycerin, as well as the nature of the feed.

Supplementation of chitosan levels did not significantly alter the in vitro digestibility of nutrients (*p* > 0.05). At 24 h of incubation, the IVNDFD and IVADFD when adding chitosan ranged from 62.33 to 64.75% DM and 50.88 to 53.83% DM, respectively. The chitosan shifted fermentation toward a pathway with greater energetic potential without adversely affecting feed digestion. This result agreed with previous studies. Goiri et al. [22] investigated the effects of supplementation at 0.36% BW of chitosan and reported no differences in OM and CP digestibility in sheep. Similarly, Del Valle et al. [27] reported that the inclusion of 0.4% chitosan in the diet had no effect on digestibility coefficients. With the supplementation of chitosan in lactating cows at 0.4% with whole raw soybeans (WRS), nutrient digestibility was not affected [9], but Paiva et al. [28] noted that chitosan provision up to 2.25% BW did not affect DM, DM, EE or NDF digestibility in lactating cows. Therefore, the results demonstrated that concentration of chitosan supplementation during 0.36 to 2.25% had no negative effect on nutrient digestion. This was also found in the present work, in which chitosan was added within the range mentioned above (2% of substrate).

### 3.5. Concentration Total Volatile Fatty Acid and Profiles

There were no interactions on total TVFA and VFA profiles between crude glycerin and chitosan (Table 5; *p* > 0.05). In addition, no significant changes (*p* > 0.05) were observed in TVFA, C2, or C4 content among treatments and between the main effects of crude glycerin with chitosan. At 4 h of incubation, ruminal C3 content and the C2 to C3 ratio changed significantly when crude glycerin and chitosan was added (*p* < 0.05). The 21% crude glycerin incorporated into TMR, in combination with 2% additional chitosan, increased C3 content by 26.41%, whereas the ratio of C2 to C3 was reduced by 31% when compared to the control group. The sample’s C3 proportion increased with higher amounts of crude glycerin in the TMR, this increase being mainly at the expense of C2. The main effect of crude glycerin inclusion (21% vs. 0% in TMR) was an 18.69% increase in C3 content and a 22.35% reduction in the C2 to C3 ratio. This likely occurred because crude glycerin, which is similar to a fermentable carbohydrate source, underwent ruminal fermentation to C3. Van Cleef et al. [6] noted that the change in the relative C3 ratio was probably due to the fermentation properties of crude glycerin, which is fermented mainly by *Selenomonas* bacteria and mostly used by ruminants during the first 4–6 h after digestion. Almeida et al. [29] found the higher ratio of crude glycerin (~43%) is quickly absorbed by the rumen wall, whereas 25% to 45% is converted to C2 and C3 by an alternative fermentative route (via succinate) and does not produce lactic acid, thus advancing the rumen’s maturation and enhancing diet use. Moreover, the C3 synthesis route is known to act as a hydrogen sink and thus might lower CH_4_ emissions, as shown in Table 3 [7].

The enhancement in the C3 concentration and the decrease in the C2 to C3 ratio in ruminal fluid are in agreement with previous studies. Avila et al. [7] found that changes to the VFA profile were observed, with C3 enhanced at the expense of C2 and C4 contents, in an in vitro study in which crude glycerin (99.5% purity) was added at up to 21%. Similarly, Rico et al. [30] reported that the fermenters in a continuous culture system with diets containing up to 8% crude glycerin can increase C3 content with a reduction in promotion of C2 to C3. In addition, Chanjula et al. [31] demonstrated that replacing corn grain with 20% of crude glycerin in the concentrated diet of growing goats increased C3 concentration by 46.52%, whereas C2 to C3 ratio was reduced by 35.53% when compared to the group fed no crude glycerin.

Propionate concentration increased by 11.75% when increasing levels of chitosan at 2% of substrate, whereas the C2 to C3 ratio depressed by 13.99% compared to the 0% chitosan group. Zanferari et al. [9] noted that chitosan may have mechanical effect similar to monensin, which was related to changes in the VFA profile, mainly decreasing C2 and enhancing C3, even at low dietary inclusion levels. Previous studies demonstrated that chitosan has persistently enhanced ruminal C3 concentration in both in vitro and in vivo experiments. In an in vitro experiment, Belanche et al. [32] found that the chitosan level was highest C3 concentration of 2 g/L. This result agreed with a study that used the Rusitec system, in which the added chitosan increased C3 concentration up to 36.8% [32]. Furthermore, the results of in vivo studies have shown that chitosan supplementation in sheep (1.36 g/kg BW; [22]), beef steers (1.5 g/kg BW; [23]), grazing steers (1.6 g/kg of concentrate; [33]) and dairy cows (4 g/kg DM; [9]) could increase ruminal the molar proportion of C3 and a reduction in C2 to C3 ratio. Thus, chitosan may be highlighted as a potential ruminal modulator to enhance VFA profiles.

## 4. Conclusions

Based on the present in vitro study, the inclusion of crude glycerin at 21% in TMR diets with chitosan supplementation at 2% enhanced ruminal propionate concentration and reduced methane production by 54% without causing any detrimental effect on gas kinetics or nutrient digestibility. However, an additional further study on a population of methanogen bacteria influenced by crude glycerin and chitosan is needed.

## Figures and Tables

**Table 1 animals-10-00037-t001:** Ingredients and chemical composition used in the total mixed ration (TMR).

Items	TMR1	TMR2	TMR3	Chitosan
Ingredients (kg dry matter; DM)				
Rice straw	30.00	30.00	30.00	
Crude glycerin ^a^	0.00	10.50	21.00	
Cassava chips	40.00	30.00	20.00	
Rice bran	7.40	6.90	6.31	
Palm kernel meal	9.00	9.00	9.00	
Soybean meal	9.00	9.00	9.00	
Molasses, liquid	1.00	1.00	1.00	
Urea	2.10	2.10	2.19	
Pure sulfur	0.50	0.50	0.50	
Mineral premix ^b^	0.50	0.50	0.50	
Salt	0.50	0.50	0.50	
Chemical composition				
Dry matter, %	92.58	92.55	92.84	98.90
Organic matter, %DM	92.92	95.12	94.20	99.73
Ash, %DM	7.08	4.88	5.80	0.27
Crude protein, %DM	14.32	14.06	14.05	0.53
Ether extract, %DM	1.03	4.82	8.19	-
Neutral detergent fiber, %DM	46.21	44.53	40.98	-
Acid detergent fiber, %DM	19.32	18.45	18.23	-
Solubility, %	-	-	-	98.70
Deacetylation degree, %	-	-	-	88.00

^a^ Crude glycerin contain 66.45% glycerol, 0.02% crude protein, 48.75% ether extract, 0.56% sodium and 4.53 methanol; ^b^ Mineral premix, per kg of premix: IU: vit. A 10,000,000, vit. E 70,000, vit. D 1,600,000; g: Fe 50, Zn 40, Mn 40, Co 0.1, Cu 10, Se 0.1, I 0.5.

**Table 2 animals-10-00037-t002:** Supplementation of glycerin with chitosan on in vitro gas kinetics and cumulative gas production at 96 h after incubation.

Treatment	Crude Glycerin (%)	Chitosan (%)	Gas Production Parameters ^a^			Cumulative Gas (mL/g DM Basis)
			*b*	*c*	*L*	
1	0	68.68	0.071	2.87	68.68	66.09
2	0	66.86	0.072	2.70	66.86	64.27
3	0	67.63	0.091	2.75	67.63	65.04
4	10.5	71.73	0.078	3.11	71.73	69.14
5	10.5	71.89	0.092	3.10	71.89	69.30
6	10.5	72.25	0.093	3.54	72.25	69.66
7	21	69.91	0.085	2.98	69.91	67.32
8	21	70.46	0.077	3.09	70.46	67.87
9	21	69.37	0.084	2.97	69.37	66.78
SEM		5.56	0.004	1.22	5.56	4.51
Main effect						
Crude glycerin						
	0		67.72	0.08	2.77	65.13
	10.5		71.96	0.09	3.25	69.36
	21		69.91	0.08	3.01	67.32
SEM			4.56	0.01	1.45	4.15
*p*-Value			0.54	0.78	0.12	0.65
Chitosan						
		0	70.11	0.08	2.99	67.51
		1	69.74	0.08	2.96	67.14
		2	69.75	0.09	3.09	67.16
SEM			5.55	0.01	1.23	4.89
*p*-Value			0.87	0.19	0.54	0.45
Interaction						
*p*-Value			0.47	0.25	0.32	0.77

^a^*b*, asymptotic gas production (mL/1 g DM); *c*, rate of gas production (/h); *L*, initial delay before gas production begins (h). SEM, standard error of the mean.

**Table 3 animals-10-00037-t003:** Influence of crude glycerin with chitosan on pH, ammonia-nitrogen (NH_3_-N) content and methane production using in vitro gas production technique after 2 and 4 h of incubation.

Treatment	Crude Glycerin (%)	Chitosan (%)	pH		NH_3_-N (mg/dL)		Methane (mL/1 g Dry Matter Substrate)	
			2 h	4 h	2 h	4 h	2 h	4 h
1	0	0	6.71	6.66	16.33	17.10	12.74	40.84
2	0	1	6.73	6.69	16.75	16.89	11.10	38.22
3	0	2	6.71	6.66	16.46	16.78	13.70	30.46
4	10.5	0	6.72	6.70	16.68	16.96	10.94	34.64
5	10.5	1	6.72	6.65	16.52	16.77	10.70	34.50
6	10.5	2	6.74	6.71	16.61	16.82	10.94	24.72
7	21	0	6.69	6.65	14.64	14.71	12.68	24.72
8	21	1	6.70	6.65	15.56	15.66	9.82	22.46
9	21	2	6.76	6.67	14.61	15.38	9.84	18.92
SEM			0.02	0.01	1.15	2.19	1.50	2.76
Main effect								
Crude glycerin								
	0		6.72	6.67	16.51	16.92	12.51	36.51
	10.5		6.73	6.69	16.60	16.85	10.86	31.29
	21		6.72	6.66	17.94	15.25	10.78	22.03
SEM			0.03	0.02	1.23	2.15	1.45	2.89
*p*-value			0.12	0.32	0.45	0.87	0.15	0.04
Chitosan								
		0	6.71	6.67	15.88	16.26	12.12	33.40
		1	6.72	6.66	16.28	16.44	10.54	31.73
		2	6.74	6.68	15.89	16.33	11.49	24.70
SEM			0.02	0.01	1.17	2.20	1.36	2.78
*p*-value			0.36	0.56	0.72	0.30	0.68	0.03
Interaction								
*p*-value			0.90	0.48	0.16	0.24	0.25	0.24

SEM, standard error of the mean.

**Table 4 animals-10-00037-t004:** Influence of crude glycerin with chitosan on in vitro digestibility of nutrients after 12 and 24 h of incubation using in vitro gas production technique.

Treatment	Crude Glycerin (%)	Chitosan (%)	IVDMD (% DM)		IVOMD (% DM)		IVNDFD (% DM)		IVADFD (% DM)	
			12 h	24 h	12 h	24 h	12 h	24 h	12 h	24 h
1	0	0	52.94	63.40	74.42	76.34	52.03	63.25	44.81	52.23
2	0	1	53.46	66.52	73.98	76.26	51.62	63.47	44.98	55.07
3	0	2	53.61	67.17	74.87	77.09	53.53	64.12	44.19	54.20
4	10.5	0	52.86	67.07	76.17	75.90	53.32	63.83	42.63	50.11
5	10.5	1	54.61	67.54	74.71	75.90	51.53	63.00	44.82	53.80
6	10.5	2	56.61	67.57	73.90	75.49	53.36	60.15	45.76	51.54
7	21	0	55.39	68.10	75.22	76.71	54.61	65.06	41.38	51.53
8	21	1	56.35	66.73	76.13	77.14	51.56	64.87	44.61	51.06
9	21	2	57.13	65.21	75.40	76.19	49.72	64.33	46.82	50.05
SEM			3.05	3.44	4.23	5.34	3.45	3.89	2.78	2.98
Main effect										
Crude glycerin										
	0		53.34	65.70	74.42	76.56	52.39	63.61	44.66	53.83
	10.5		54.69	67.39	74.93	75.76	52.74	62.33	44.40	51.82
	21		56.29	66.68	75.58	76.68	51.96	64.75	44.27	50.88
SEM			3.10	3.21	4.12	5.31	3.25	3.77	2.45	2.88
*p*-Value			0.09	0.11	0.54	0.36	0.85	0.45	0.12	0.09
Chitosan										
		0	53.73	66.19	75.27	76.32	53.32	64.05	42.94	51.29
		1	54.81	66.93	74.94	76.43	51.57	63.78	44.80	53.31
		2	55.78	66.65	74.72	76.26	52.20	62.87	45.59	51.93
SEM			3.00	3.12	3.98	5.01	3.01	3.55	2.13	2.08
*p*-Value			0.11	0.84	0.23	0.77	0.62	0.12	0.33	0.88
Interaction										
*p*-Value			0.12	0.31	0.55	0.62	0.74	0.12	0.65	0.22

SEM, standard error of the mean; IVDMD, in vitro dry matter digestibility, IVOMD, in vitro organic matter digestibility, IVNDFD, in vitro neutral detergent fiber digestibility; IVADFD, in vitro acid detergent fiber digestibility.

**Table 5 animals-10-00037-t005:** Influence of crude glycerin with chitosan on in vitro volatile fatty acids (VFAs) profiles after 2 and 4 h of incubation.

Treatment	Crude Glycerin (%)	Chitosan (%)	Acetate (%)		Propionate (%)		Butyrate (%)		Acetate to Propionate Ratio		Total (mmol/L)	
			2 h	4 h	2 h	4 h	2 h	4 h	2 h	4 h	2 h	4 h
1	0	0	65.27	64.97	20.13	20.79	14.60	14.24	3.24	3.13	85.65	94.43
2	0	1	65.32	64.02	20.18	21.23	14.50	14.75	2.70	3.02	86.11	96.36
3	0	2	65.59	65.79	21.11	23.45	13.30	10.76	2.22	2.81	86.26	93.54
4	10.5	0	65.41	65.79	21.03	22.12	13.56	12.09	2.53	2.97	84.90	95.52
5	10.5	1	64.25	64.82	20.16	24.70	15.59	10.48	2.53	2.62	86.28	96.90
6	10.5	2	64.29	62.08	21.17	26.29	14.54	11.63	2.47	2.36	84.85	94.53
7	21	0	64.23	62.58	21.70	25.92	14.07	11.50	2.34	2.41	86.12	96.96
8	21	1	63.33	62.62	22.12	26.35	14.55	11.03	2.21	2.38	88.75	96.16
9	21	2	63.93	60.90	22.25	28.25	13.82	10.85	2.80	2.16	87.88	97.06
SEM			5.43	6.45	1.53	1.55	1.31	1.48	0.15	0.14	6.99	7.87
Main effect												
Crude glycerin												
	0		65.39	64.93	20.47	21.82	14.13	13.25	2.72	2.99	86.01	94.78
	10.5		64.65	64.23	20.79	24.37	14.56	11.40	2.51	2.65	85.34	95.65
	21		63.83	62.03	22.02	26.84	14.15	11.13	2.45	2.32	87.58	96.73
SEM			5.33	6.66	1.51	1.53	1.33	1.51	0.16	0.13	7.15	8.98
*p*-Value			0.99	0.58	0.25	0.03	0.55	0.45	0.12	0.02	0.22	0.34
Chitosan												
		0	64.97	64.45	20.95	22.94	14.08	12.61	2.70	2.84	85.56	95.64
		1	64.30	63.82	20.82	24.09	14.88	12.09	2.48	2.67	87.05	96.47
		2	64.60	62.92	21.51	26.00	13.89	11.08	2.50	2.44	86.33	95.04
SEM			5.47	6.99	1.49	1.50	1.25	1.56	0.14	0.12	6.88	7.25
*p*-Value			0.11	0.35	0.19	0.05	0.18	0.81	0.87	0.04	0.64	0.25
Interaction												
*p*-Value			0.33	0.09	0.11	0.84	0.45	0.54	0.33	0.32	0.45	0.77

SEM, standard error of the mean

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
