# Peer review of "Combining Crude Glycerin with Chitosan Can Manipulate In Vitro Ruminal Efficiency and Inhibit Methane Synthesis"

_animals, 2019, doi:10.3390/ani10010037_

Round 1

Reviewer 1 Report

The manuscript reports the evaluation of glycerol and chitosan inclusion in a TMR diet on CH4 production, rumen fermentation and digestibility. At the beginning I found the study methodologically sound, however I have a few major concerns that to my opinion need to be addressed prior being suitable for publication. 

Major Concerns: 

1.- My main concern is that VFA’s and CH4 were measured just at 2 and 4h of incubation, and not at the end of the fermentation. So, their conclusions might be erroneous as CH4 concentrations/production could have been equalized throughout the fermentation process: This must be at least be stated in the discussion. A refer that CH4 is reduced in the early fermentation.   

2.- For CH4, NH3-N, pH, VFA’s and digestibilities authors have 2 incubation times, which to my opinion must be incorporated in the statistical model as a repeated measure

3.- When having a look at GP data with negatives values for a, it suggests that authors should have used a model with a lag phase. I think authors need to model the data, including that lag phase

4.- Gas production and CH4 data must be reported on a 1g basis, in order to be more comparable with other studies

5.- For all tables, the Glycerin*chitosan interaction was reported, however none of the evaluated variables was significant. Thus I recommend to report all tables with means of main effects separately and the p-values and sem for glycerin and chitosan and p-value for the interaction, and state in the results section that the interactions were not significant.

6.- I feel that the language must be improved, I tried to help with some changes, but think it needs a deeper revision

Minor comments

L19: Please define C3 at first mention

L21: replace “were” by “was” and delete “to”

L22: add “arrangement of treatments”

L32: mg% is the unit correct?

L40: acid”s”

L41: propoionate (C3) should have been defined in line 19, and here you can use C3

L45: Do not start a sentence with an abbreviation, please change C3 for propionate

L53: please delete CH4, it was already defined

L54: Do not start with an abbreviation

L58: please delete C3, it was already defined

L58-59: This sentence is a little bit unclear, please rephrase. I would say something like that “propionate serves as an H sink in the rumen, thus an increase in its concentration would reduce H availability and therefore CH4 production”.

L62: Please first mention that van Cleef et al indicated the possibility of reducing CH4, and the start mentioning the negative effects.

M&M

The ingredients of the TMR and their proportions (control diet) must be described

L89: Delete “in”

L90: delete “be”

L93: I would say residue instead of remainder

L104: as rumen fluid donnors

L107-108: ……. was mixed at equal proportions. 

L113: replace “were” by “was” and delete “to”

L114-115: treatments are the combination of the 3 levels of glycerin and 3 levels of chitosan.

L124: A digestibility evaluation was arranged ……..

L122: Replace “The” by “A set of”

L127: Chemical analyses instead of assay

L132-133: Check spelling. I recommend “…. was analyzed for CP and OM with the similar protocol mentioned above”………

L138: was analyzed, instead of was investigated

L139: just use C2, C3 and C4

L142-145: How did authors correct for the microbial mass in the residue?

L145: Methane, do not start the sentence with an abbreviation 

L150: Statistical analysis and gas kinetics

L151:was used for gas kinetics analysis.

L163: … was increased ……

L168-169: at what level of inclusion?

L175: which indicates ….. has a potential for animal supplementation use

L178: When having a look at GP data with negatives values for a, it suggests that authors should have used a model with a lag phase. I think authors need to model the data, including that lag phase. Besides, as the interaction was not significant, authors should report main effects instead of the interaction. GP data should be standardized on a 1 g/basis

L179: in terms

L193: The differences among trials should be discussed

L200: it is a buffered medium, so it is expected that pH levels are within those values. So please state that: Considering it is a buffered medium ………

L201: mg% unit for NH3-N, I think mg/100ml is better

L209: concentrations of CH4 could “be reduced”

L285: …… glycerin “incorporated” into TMR ……

L287-289: Please double check this sentence, How can be C3 at the expense of itself?

L292: van Cleef with uppercase (Van Cleef), since it is at the beginning of a sentence

L311: a mechanical “effect”?

Sincerely, 

The reviewer

Author Response

Author's Reply to the Review Report (Reviewer 1)

The manuscript reports the evaluation of glycerol and chitosan inclusion in a TMR diet on CH4 production, rumen fermentation and digestibility. At the beginning I found the study methodologically sound, however I have a few major concerns that to my opinion need to be addressed prior being suitable for publication.

Response: We appreciate thanks so much for comments and suggestion made on manuscript. Above all, the authors felt that all points made were very useful and have incorporated most of the corrections where necessary as suggested in order to make the manuscript ready for possible publication. All those corrected and modified appear in track change mode in the manuscript. Please see information given by the authors following the suggestions and comments made by all reviewers. Thanks!

Major Concerns:

1.- My main concern is that VFA’s and CH4 were measured just at 2 and 4h of incubation, and not at the end of the fermentation. So, their conclusions might be erroneous as CH4 concentrations/production could have been equalized throughout the fermentation process: This must be at least be stated in the discussion. A refer that CH4 is reduced in the early fermentation.  

Response: Thank you for the nice suggestion and now we have added into the text as “However, present study found CH4 was reduced in the early fermentation of 4 h of incubation, thus throughout the fermentation process (96 h of incubation) should be further elucidated.” Please see in section of “In vitro rumen parameters and CH4 concentration” in page 9.

2.- For CH4, NH3-N, pH, VFA’s and digestibilities authors have 2 incubation times, which to my opinion must be incorporated in the statistical model as a repeated measure

Response: We have re-calculated and provided detail in section of “Statistical analysis and gas kinetics” as “Data from 2 incubation times of CH4, NH3-N, pH, VFA’s (2 and 4 h of incubation) and digestibilities (12 and 24 h of incubation) were analyzed as a mixed model with repeated measurements using the PROC MIXED of SAS [17], in which the effect of sampling time was considered fixed and that of the inoculum was considered random. Data of in vitro gas kinetics and cumulative gas production were subjected to the General Linear Models (GLM) procedures of SAS [17]. Differences among treatment means for all parameters were contrasted by Tukey’s Multiple Comparison Test.” Please see in manuscript.

3.- When having a look at GP data with negatives values for a, it suggests that authors should have used a model with a lag phase. I think authors need to model the data, including that lag phase

Response: We have already re-analyzed regarding to your comments by choose the model of France et al. (2000; Br J Nutr, 83:143–150.) which including lag phase the following model:

Gas production = b × [1 – e-c(t-L)]

where b is the asymptotic gas production (ml/ 1 g DM); c is the rate of gas production (/h) and L (h) is the discrete lag time prior gas production.”. Furthermore,   results and discussion as well as data kinetic of gas in table were also revised. Please see in manuscript.

4.- Gas production and CH4 data must be reported on a 1g basis, in order to be more comparable with other studies

Response: Thanks. Now, we have re-calculated for gas production and CH4 data and reported on a 1 g basis, in order to be more comparable with other studies. Please see in the tables and throughout manuscript.

5.- For all tables, the Glycerin*chitosan interaction was reported, however none of the evaluated variables was significant. Thus I recommend to report all tables with means of main effects separately and the p-values and sem for glycerin and chitosan and p-value for the interaction, and state in the results section that the interactions were not significant.

Response: It is very great suggestion and we have all agreed with. Now, all Tables have been modified accordingly to your comment already. In addition, results and discussion was also revised wherever the Table changed. Please see in manuscript.

6.- I feel that the language must be improved, I tried to help with some changes, but think it needs a deeper revision

Response: Thanks so much for your comments and suggestion us to improving English. Now, we have revised accordingly your comments point-by-point as well as wherever it need to improve. Furthermore, before we submit manuscript to the journal, we have send our Manuscript to the company namely Papercheck (https://www.papercheck.com/) located in California, United States for English grammar edited already (Order no. 775639). Herein, we have attached receipt of payment for English grammar proof. Therefore, we think that throughout manuscript have been improved in English language.

Minor comments

L19: Please define C3 at first mention

Response: We have already modified. Please see in manuscript.

L21: replace “were” by “was” and delete “to”

Response: We have already modified. Please see in manuscript.

L22: add “arrangement of treatments”

Response: We have already modified. Please see in manuscript.

L32: mg% is the unit correct?

Response: We have changed to mg/dl throughout the manuscript.

L40: acid”s”

Response: We have already modified. Please see in manuscript.

L41: propoionate (C3) should have been defined in line 19, and here you can use C3

Response: We have already modified. Please see in manuscript.

L45: Do not start a sentence with an abbreviation, please change C3 for propionate

Response: We have already modified. Please see in manuscript.

L53: please delete CH4, it was already defined

Response: We have already modified. Please see in manuscript.

L54: Do not start with an abbreviation

Response: We have already modified. Please see in manuscript.

L58: please delete C3, it was already defined

Response: We have already modified. Please see in manuscript.

L58-59: This sentence is a little bit unclear, please rephrase. I would say something like that “propionate serves as an H sink in the rumen, thus an increase in its concentration would reduce H availability and therefore CH4 production”.

Response: We have already modified. Please see in manuscript.

L62: Please first mention that van Cleef et al indicated the possibility of reducing CH4, and the start mentioning the negative effects.

Response: We have already modified. Please see in manuscript.

M&M

The ingredients of the TMR and their proportions (control diet) must be described

Response: We have already described. Please see in manuscript.

L89: Delete “in”

Response: We have already modified. Please see in manuscript.

L90: delete “be”

Response: We have already modified. Please see in manuscript.

L93: I would say residue instead of remainder

Response:

L104: as rumen fluid donnors

Response: We have already modified. Please see in manuscript.

L107-108: ……. was mixed at equal proportions.

Response: We have already modified. Please see in manuscript.

L113: replace “were” by “was” and delete “to”

Response: We have already modified. Please see in manuscript.

L114-115: treatments are the combination of the 3 levels of glycerin and 3 levels of chitosan.

Response: We have already modified. Please see in manuscript.

L124: A digestibility evaluation was arranged ……..

Response: We have already modified. Please see in manuscript.

L122: Replace “The” by “A set of”

Response: We have already modified. Please see in manuscript.

L127: Chemical analyses instead of assay

Response: We have already modified. Please see in manuscript.

L132-133: Check spelling. I recommend “…. was analyzed for CP and OM with the similar protocol mentioned above”………

Response: We have already modified. Please see in manuscript.

L138: was analyzed, instead of was investigated

Response: We have already modified. Please see in manuscript.

L139: just use C2, C3 and C4

Response: We have already modified. Please see in manuscript.

L142-145: How did authors correct for the microbial mass in the residue?

Response: In our opinion, microbial mass in the residue is very low and did not much affect content of fiber determination. Furthermore, during content was filtered through pre-weighed Gooch crucibles (40 mm of porosity), microbial mass might be washed out from the samples already. As, this methodology have been accepted in many International Standard Journal.

L145: Methane, do not start the sentence with an abbreviation

Response: We have already modified. Please see in manuscript.

L150: Statistical analysis and gas kinetics

Response: We have already modified. Please see in manuscript.

L151:was used for gas kinetics analysis.

Response: We have already modified. Please see in manuscript.

L163: … was increased ……

Response: We have already modified. Please see in manuscript.

L168-169: at what level of inclusion?

Response: We have modified to “Crude glycerin contain low methanol at 4.53% DM and was lower than the levels reported by Lage et al. [18], who demonstrated that 8.7% of methanol in crude glycerin did not adversely affect the health status of feedlot lambs fed crude glycerin 12% on DM basis of concentrate diet.”

L175: which indicates ….. has a potential for animal supplementation use

Response: We have already modified. Please see in manuscript.

L178: When having a look at GP data with negatives values for a, it suggests that authors should have used a model with a lag phase. I think authors need to model the data, including that lag phase. Besides, as the interaction was not significant, authors should report main effects instead of the interaction. GP data should be standardized on a 1 g/basis

Response: We have already re-analyzed regarding to your comments by choose the model of France et al. (2000; Br J Nutr, 83:143–150.) which including lag phase the following model:

Gas production = b × [1 – e-c(t-L)]

where b is the asymptotic gas production (ml/ 1 g DM); c is the rate of gas production (/h) and L (h) is the discrete lag time prior gas production.”. Furthermore,   results and discussion as well as data kinetic of gas in table were also revised. In addition, we have re-calculated for gas production and CH4 data and reported on a 1 g basis, in order to be more comparable with other studies. Please see in the tables and throughout manuscript.

L179: in terms

Response: We have already modified. Please see in manuscript.

L193: The differences among trials should be discussed

Response: We have already discussed. Please see in manuscript.

L200: it is a buffered medium, so it is expected that pH levels are within those values. So please state that: Considering it is a buffered medium ………

Response: We have already modified. Please see in manuscript.

L201: mg% unit for NH3-N, I think mg/100ml is better

Response: We have change to mg/dl.

L209: concentrations of CH4 could “be reduced”

Response: We have already modified. Please see in manuscript.

L285: …… glycerin “incorporated” into TMR ……

Response: We have already modified. Please see in manuscript.

L287-289: Please double check this sentence, How can be C3 at the expense of itself?

Response: We have modified to “The sample’s C3 proportion increased with higher amounts of crude glycerin in the TMR, this increase being mainly at the expense of C2.”

L292: van Cleef with uppercase (Van Cleef), since it is at the beginning of a sentence

Response: We have already modified. Please see in manuscript.

L311: a mechanical “effect”?

Response: We have already modified. Please see in manuscript.

Reviewer 2 Report

Lines 103 - 105 - What ration were these animals fed (high forage, mixed forage/grain, etc.)? This needs to be defined as the type of ration alters the microbial community. Thus, in vitro results my differ due to rations fed to the donor animals. Just defining the ration fed will be helpful to  readers.

Lines 204-207 - Any explanation as to why there was no change in CH4 at the 2 hour incubation period?

Lines 327-328 - You may want to expand your comments on the need for in vivo work. This is a critical step in determining the potential of compounds to lower CH4 production. It should be also pointed out that these in vivo trials need to be long-term to assure that the rumen microorganisms don't adapt to the compound.

Author Response

Author's Reply to the Review Report (Reviewer 2)

Lines 103 - 105 - What ration were these animals fed (high forage, mixed forage/grain, etc.)? This needs to be defined as the type of ration alters the microbial community. Thus, in vitro results my differ due to rations fed to the donor animals. Just defining the ration fed will be helpful to  readers.

Response: We have defined in manuscript as “The animals were fed with TMR1 (control diet; Table 1) at 2.5% BW daily.” Please see in Page 5.

Lines 204-207 - Any explanation as to why there was no change in CH4 at the 2 hour incubation period?

Response: We have already explained as “Concentration of CH4 production was not changed after incubation at 2 h, which might be due to the initial ruminal fermentation process, thus lowering substrate for CH4 to synthesis. “ Please see in section of “In vitro rumen parameters and CH4 concentration” in page 9.

Lines 327-328 - You may want to expand your comments on the need for in vivo work. This is a critical step in determining the potential of compounds to lower CH4 production. It should be also pointed out that these in vivo trials need to be long-term to assure that the rumen microorganisms don't adapt to the compound.

Response: Thanks for suggestion and now we have modified as “However, an additional further study on the population of methanogen bacteria as influenced of crude glycerin and chitosan is needed.” Please see in section of Conclusion page 14.

Thank you very much!

Round 2

Reviewer 1 Report

I would like to congratulate the authors for their effort. I believe that the current version is much closer for being suitable for publication. 

I have a few more comments: 

L265-266: ……. Thus “the inhibitory effect of glycerin and chitosan” throughout the “complete” fermentation process ……

For tables:

I would place the p-value for the interaction below its data. I think that data would be more easy to follow if you put independent variables on columns and the dependent ones on the rows Table 2 and 3. If you included sampling time as a repeated measure you should include its p-value and the glycerin*sampling time, chitosan*sampling time and glycerin*chitosan*sampling time interactions

P-value

Glycerin

SEM

p-Value

Chitosan

SEM

G*Ch

G*IT

Ch*IT

G*Ch*IT

0

10,5

21

0

1

2

CH4

Best regards

The reviewer